# Serum GlycA Level Is Elevated in Active Systemic Lupus Erythematosus and Correlates to Disease Activity and Lupus Nephritis Severity

**DOI:** 10.3390/jcm9040970

**Published:** 2020-03-31

**Authors:** Tim Dierckx, Laurent Chiche, Laurent Daniel, Bernard Lauwerys, Johan Van Weyenbergh, Noémie Jourde-Chiche

**Affiliations:** 1Rega Institute, Laboratory of Clinical and Epidemiological Virology, Department of Microbiology and Immunology, KU Leuven, 3000 Leuven, Belgium; tim.dierckx@kuleuven.be (T.D.); johan.vanweyenbergh@kuleuven.be (J.V.W.); 2Médecine Interne, Hôpital Européen, 13003 Marseille, France; l.chiche@hopital-europeen.fr; 3Hôpital de la Timone, Marseille, Laboratoire d’Anatomie Pathologique, AP-HM, 13005 Marseille, France; laurent.daniel@ap-hm.fr; 4C2VN, INRA 1260, INSERM 1263, Aix-Marseille Université, 13005 Marseille, France; 5Institut de Recherches Expérimentales et Cliniques, Université catholique de Louvain, 1200 Brussels, Belgium; bernard.lauwerys@uclouvain.be; 6Department of Rheumatology, Cliniques Universitaires Saint-Luc, 1200 Brussels, Belgium; 7Hôpital de la Conception, Centre de Néphrologie et Transplantation Rénale, AP-HM, 13005 Marseille, France

**Keywords:** systemic lupus erythematosus, disease activity, lupus nephritis severity, glycoprotein acetylation, systemic chronic inflammation

## Abstract

Objective: Reliable non-invasive biomarkers are needed to assess disease activity and prognosis in patients with systemic lupus erythematosus (SLE). Glycoprotein acetylation (GlycA), a novel biomarker for chronic inflammation, has been reported to be increased in several inflammatory diseases. We investigated the relevance of serum GlycA in SLE patients exhibiting various levels of activity and severity, especially with regards to renal involvement. Methods: Serum GlycA was measured by nuclear magnetic resonance spectroscopy in samples from well characterized SLE patients and from both healthy controls and patients with other kidney diseases (KD). Disease activity was evaluated using the Systemic Lupus Erythematosus Activity Index 2000 (SLEDAI-2K). Renal severity was assessed by kidney biopsy. Results: Serum GlycA was elevated in active (*n* = 105) compared to quiescent SLE patients (*n* = 39, *p* < 10^−6^), healthy controls (*n* = 20, *p* = 0.009) and KD controls (*n* = 21, *p* = 0.04), despite a more severely altered renal function in the latter. GlycA level was correlated to disease activity (SLEDAI-2K, ρ = 0.37, *p* < 10^−4^), C-reactive protein, neutrophil count, triglyceride levels, proteinuria and inversely to serum albumin. In patients with biopsy-proven lupus nephritis (LN), GlycA levels were higher in proliferative (*n* = 26) than non-proliferative LN (*n* = 10) in univariate analysis (*p* = 0.04), and was shown to predict proliferative LN independently of renal parameters, immunological activity, neutrophil count and daily corticosteroid dosage by multivariate analysis (*p* < 5 × 10^−3^ for all models). In LN patients with repeated longitudinal GlycA measurement (*n* = 11), GlycA varied over time and seemed to peak at the time of the flare. Conclusions: GlycA, as a summary measure for different inflammatory processes, could be a valuable biomarker of disease activity in patients with SLE, and a non-invasive biomarker of pathological severity in the context of LN.

## 1. Introduction

Reliable and non-invasive biomarkers are still needed in systemic lupus erythematosus (SLE) to assess disease activity, severity of organ involvement, and long-term prognosis [1]. Lupus nephritis (LN) is one of the most frequent complications of SLE [2] and is associated with increased cardiovascular morbidity [3] and mortality [4], in addition to the risk of chronic kidney disease (CKD) and end-stage renal disease [5]. The severity of LN, which conditions the intensity of treatment, is defined by the presence of active proliferative inflammatory lesions on kidney biopsy [2,6,7]. Although most manifestations of SLE including nephritis are inflammatory [8], C-reactive protein (CRP) elevation is more related to infections than to disease activity in SLE [9]. 

Glycoprotein acetylation (GlycA) is a novel nuclear magnetic resonance (NMR) spectroscopy-derived biomarker of systemic inflammation. GlycA signals, measured in serum or plasma, quantify the glycosylation level of several acute-phase proteins (predominantly α1-acid glycoprotein, haptoglobin, α1antitrypsin, α1antichymotrypsin and transferrin [10,11]), as a consequence of inflammatory stimuli. The GlycA signal can be interpreted as a biomarker of systemic inflammation [12], summarizing the activity of multiple inflammatory pathways [13]. GlycA has been shown to be associated with the risk of cardiovascular events, cardiovascular mortality and all-cause mortality in the general population, independently of CRP [14,15,16]. A large cohort study has additionally shown independent associations to estimated glomerular filtration rate (eGFR) and albuminuria [17]. We recently showed that GlycA could be a marker of disease activity in inflammatory bowel disease in patients without CRP elevation [18]. Moreover, large scale gene correlation network analysis has shown that GlycA can be associated to neutrophil gene expression [16], which has been shown to be associated with LN occurrence and severity in SLE patients [19,20]. 

Here, we investigated if GlycA could be associated with SLE disease activity and LN severity in a large cohort of SLE patients including patients with biopsy-proven LN, as well as appropriate control populations.

## 2. Materials and Methods

### 2.1. Patient Demographics & Ethics

This study was conducted in accordance with the principles of the declaration of Helsinki. Secondary analysis of data was approved by the Medical Ethics Commission of University Hospitals Leuven, Belgium (s57931). 

Patients with biopsy-proven LN, and patients with other kidney diseases (KD), were included from the biobank DC-2012-1704, approved by the French Ministry of Health, in the Hôpital de la Conception, AP-HM, Marseille, France. All patients gave written informed consent before any procedure.

Other samples from patients with SLE were obtained from patients recruited in the LOUvain Lupus Nephritis InCeption (LOULUNIC) cohort, and patients followed-up at the Lupus Clinic of the Université Catholique de Louvain (UCL), Brussels, Belgium, as were samples from age- and sex-matched healthy controls. All patients and controls gave written informed consent before serum samplings. All patients with SLE responded to the SLICC 2012 classification criteria [21].

### 2.2. Clinical and Conventional Biological Metrics

A graphical overview of all assayed samples can be found in Appendix A. Samples were grouped as either originating from healthy controls, from patients with various KD (comprising patients with membranous nephropathy, IgA nephropathy, diabetic kidney disease or hypertensive nephropathy) or from SLE patients. SLE patients were further subclassified as clinically quiescent (defined as SLEDAI-2K ≤ 4 without clinical activity, with or without maintenance therapy, regardless of immunological activity [22]), or active SLE patients. Among active SLE patients, some were sampled at the time of a biopsy-proven LN, and showed either severe (“proliferative”) nephritis (defined as class III or IV +/− V, with active lesions, per the ISN/RPS 2003 classification) or milder (“non-proliferative”) nephritis (defined as class I, II or isolated class V). In proliferative LN, the percentages of active and chronic lesions per the ISN/RPS 2003 classification were gathered. Detailed demographical and clinical parameters of these groups can be found in Appendix A. Longitudinal samples comprising a sample taken during a flare of LN were only available for a subgroup of patients.

Age, gender, ethnicity, BMI, smoking status, overall lupus activity (Systemic Lupus Erythematosus Activity Index 2000, SLEDAI-2K) [23], serum CRP level, serum albumin and creatinine levels, estimated glomerular filtration rate (eGFR, calculated with the MDRD equation [24]), C3 and C4 concentration, urinary protein/creatinine ratio (UPCR), presence of anti-dsDNA antibodies as a binary variable (above 16 UI/mL by ELIA^TM^, ThermoFisher, Cambridge, MA, USA, or above 10 IU/mL using the Farr assay from Trinity Biotech, Bray, Ireland), anti-dsDNA levels by ELIA^TM^, total cholesterol, LDL and HDL cholesterol, triglyceride levels, daily glucocorticoid dosage, and hydroxychloroquine prescription were tested for association to GlycA.

### 2.3. GlycA Quantification

GlycA concentration was quantified using the Nightingale Health Ltd. high-throughput metabolomics platform (Helsinki, Finland) [10,11]. Briefly, a ^1^H-NMR spectrum is taken from each sample, with the area under the peak measured at approximately 2 ppm quantifying signal originating from N-acetyl sugar groups found on a heterogeneous set of acute phase glycoproteins, including α-1-acid glycoprotein, α-1-antitrypsin, α-1-antichymotryspin, haptoglobin and transferrin. From the NMR spectrum, over 220 metabolite measures can be quantified. From these, an *a priori* selection of metabolites was made based on previous reports [25,26,27,28] (i.e., total cholesterol and triglyceride concentration, HDL and LDL cholesterol concentration, albumin and creatinine levels), as the available sample size did not support full discovery-oriented analysis. The full metabolite measurements are made available in the Appendix A. Laboratory and NMR measurements were found to be highly correlated for creatinine and albumin concentrations (*ρ* = 0.94 and *ρ* = 0.74, respectively, with *p*-values < 10^−8^) (Appendix A).

### 2.4. Statistical Analysis

Cohort characteristics are presented as median and inter-quartile range (IQR) for continuous variables, and frequency (percentage) for categorical variables. Statistical significance of the differences between two groups was determined with Mann-Whitney-Wilcoxon’s U or Chi^2^ test, as appropriate. Spearman correlations were used to examine the association between GlycA concentration and other continuous parameters. Multivariate logistic regression on samples taken at time of renal biopsy of flaring LN patients was performed on standardized continuous variables. All statistical analysis was performed in R [29]. Figures were generated using the ggplot2 R package [30].

## 3. Results

### 3.1. GlycA is Elevated in Patients with Active SLE

GlycA was not different between healthy controls (*n* = 20) and the complete cohort of active and quiescent SLE patients (*n* = 144) (Table 1). However, patients with active SLE (*n* = 105) displayed significantly higher GlycA concentrations than healthy controls (*p* = 0.009), and quiescent SLE patients (*n* = 39, *p* < 10^−6^). Patients with active SLE also displayed higher GlycA levels than non-lupus KD controls (*n* = 21, *p* = 0.04), despite comparable CRP levels, and despite a more altered renal function in KD controls (*p* < 10^−3^) (Figure 1). NMR measures and comparisons of these NMR measures between HC, KD controls, quiescent SLE and active SLE are summarized in Table 1. 

### 3.2. GlycA Is Associated with Disease Activity in SLE

SLEDAI-2K was correlated to GlycA when considering all samples from patients with SLE (*n* = 144, ρ = 0.37, *p* < 10^−4^) (Figure 2), but not when exclusively considering samples from quiescent SLE patients (*n* = 39), or when exclusively considering samples from active SLE patients (*n* = 105). GlycA correlated well with CRP and serum creatinine levels (Figure 2), both in quiescent and active SLE patients (ρ = 0.57 and 0.36, both *p* values <10^−3^ and ρ = 0.45 and 0.25, *p* values <0.01 and 0.013, for CRP and creatinine respectively). Full results on GlycA associations with all available demographic and clinical parameters are listed in Table 2. 

Figure 3 shows GlycA levels from longitudinal samples of 11 LN patients for whom a sample was taken during a flare-up in disease activity. While insufficient observations were available to perform robust longitudinal analysis, we note that GlycA is variable over the examined time period and could be related to the flare status of the patients.

### 3.3. GlycA Is Associated with the Pathological Severity of LN

In serum samples taken at the time of renal biopsy in patients with a flare of LN (*n* = 36), GlycA concentrations were higher in patients with proliferative LN than in patients with non-proliferative LN (*p* = 0.04), despite comparable levels of eGFR and proteinuria (Figure 4). The only other parameter that showed a statistically significant difference between patients with proliferative versus non-proliferative LN in this cohort was BMI, which was lower in patients with proliferative LN (*p* = 0.04) (Table 3). GlycA levels were not associated with the percentages of activity (ρ = 0.02, *p* = 0.94) or chronicity (ρ = 0.28, *p* = 0.34) of the ISN/RPS classification, but were correlated with serum creatinine levels (ρ = 0.34, *p* = 0.04) and eGFR (ρ = −0.40, *p* = 0.02). In these 36 samples from LN patients taken at time of renal biopsy, we observed strong trends for association between GlycA and Neutrophil count (ρ = 0.35, *p* = 0.06) and daily corticosteroid dosage (ρ = −0.30, *p* = 0.09). GlycA additionally showed a strong correlation with total serum triglyceride concentration (ρ = 0.81, *p* < 10^−8^), and with total cholesterol (ρ = 0.45 *p* < 0.01) and LDL-cholesterol (ρ = 0.45, *p* < 0.01) levels (Figure 5). However, GlycA’s association to proliferative status was found to be independent from all tested clinical, immunological and NMR parameters in BMI corrected multivariate logistic regression analysis (Figure 6, full model parameters in Appendix A).

## 4. Discussion

This work reports on the relevance of GlycA measurement in the largest cohort of SLE patients tested to date [25,31]. We investigated serum GlycA level’s potential as a biomarker of activity and severity in SLE by comparing NMR measurements in samples from patients with quiescent SLE, active SLE, and control samples from both healthy and KD controls. While previous reports showed elevated GlycA in SLE patients compared to healthy controls [25,31], we extend previous findings by showing that GlycA levels are elevated in active SLE patients, and correlate with disease activity evaluated by SLEDAI-2K [25,27], provided that sufficient variation in disease activity is present in the observed population [31]. In a subset of longitudinal samples from patients with LN, we show that GlycA can vary over time and seems to increase at the time of a flare.

To our knowledge, this is the first study that had access to serum samples taken at the time of renal biopsy to determine GlycA’s association to the severity of pathological lesions. We show that GlycA levels are higher in patients with proliferative LN than in patients with non-proliferative LN. This increase could not be attributed solely to altered renal parameters: first, proteinuria levels were comparable between patients with proliferative and non-proliferative LN. Second, KD controls simultaneously displayed more altered eGFR (which is associated with higher GlycA in the general population and in the present cohort), older age (which is associated with higher GlycA in the general population) and yet lower GlycA levels than patients with LN. Third, though GlycA showed significant inverse correlation with eGFR (ρ = −0.40, *p* = 0.02) as expected [17], its association to proliferative status was found to be statistically independent of eGFR. We show that GlycA’s association to proliferative status is independent of the possible confounding factors to GlycA signal, whether they were previously described (BMI [13,32,33,34] and daily corticosteroid dosage [27]) or observed in this work (Neutrophil count, Table 3). 

Increased serum triglyceride levels have previously been shown to be related to SLE disease activity [26]. While triglyceride levels were not found to be univariately associated with LN severity in the present study, they were associated to GlycA levels in patients with LN (Table 3). 

Concerning the role of neutrophils, we have previously shown the involvement of a set of neutrophil activation related genes with disease severity in LN [20] and large-scale gene correlation network analysis has shown that GlycA can also be associated to neutrophil gene expression [16]. Here, our results show a trend towards significance for the association between neutrophil count and GlycA levels.

The significant association between BMI and the proliferative status of LN in this cohort is also worth noting (Table 3). Although obesity is not a risk factor for the development of SLE [35], the prevalence of obesity in patients with SLE is high, and higher BMI has been associated with SLE disease activity [36], chronic inflammation and cardiovascular burden [37]. Higher BMI has also been associated with an increased risk of proteinuria in SLE patients [38], but this may either reflect active LN, or be the consequence of glomerular hyperfiltration and glomerulosclerosis induced by obesity itself [39]. The correlation of GlycA with BMI is well documented [13,32,33,34] and the fact that patients with proliferative LN examined in this work displayed higher GlycA levels despite significantly lower BMIs reinforces the importance of the association between GlycA and LN severity.

We point out specific limitations and strengths of this study. The limited sample sizes and unbalanced study design inherent to convenience sampling restrict the hypotheses which can be statistically tested in this dataset. Sample sizes (*n* = 10 proliferative and *n* = 26 non-proliferative) preclude extensive logistic regression analysis comparing model sensitivity and specificity, and algorithmic identification of the best performing models. Although erythrocyte sedimentation rate can be an interesting marker in SLE, it is influenced by age, gender, anemia and renal disease, and was not available in the present cohort. Additionally, while our observations hint that GlycA could be a candidate biomarker for treatment follow-up, the lack of rigorous periodic follow-up sampling leading up to and following the flare event obfuscate the dynamics of the GlycA marker in this context. The strengths of this study lie in the availability of serum samples drawn at the time of a biopsy-proven LN flare, the access to extensive clinical measurements and the availability of robust phenotypical characterization for the examined SLE patients and biopsy-proven KD controls. 

The GlycA biomarker quantifies the degree to which specific acute-phase glycoproteins are acetylated. Further research is required to elucidate whether the observed GlycA associations can be ascribed to either the concentrations of these proteins, their glycosylation or their acetylation profiles. Interestingly, of all 220+ measures quantified in the NMR experiments, serum triglyceride levels are exceptionally well correlated with GlycA levels, both in our results and in other reports [15,17,31,40]. 

## 5. Conclusions

Serum GlycA concentration, as a summary measure for multiple inflammatory processes, could be a valuable biomarker of disease activity in patients with SLE, and of pathological severity in patients with LN. Given its strong correlation with triglycerides and cholesterol in this study, GlycA may also be a biomarker of interest for the study of accelerated atherosclerosis in SLE.

## Figures and Tables

**Figure 1 jcm-09-00970-f001:**
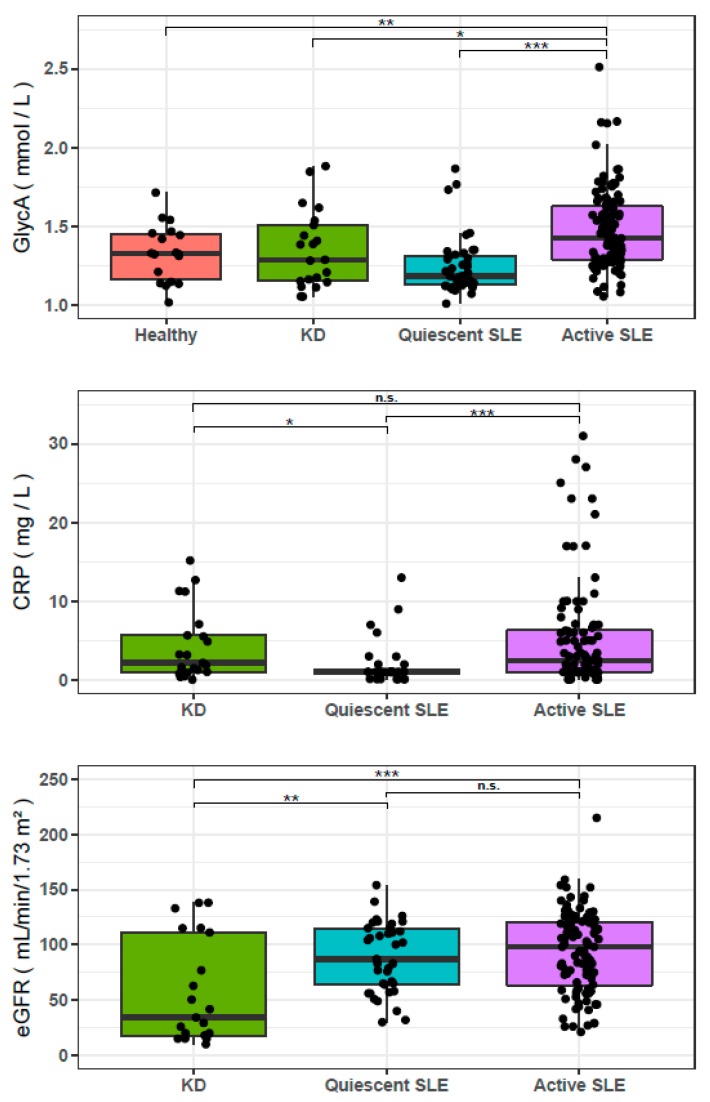
Glycoprotein Acetylation (GlycA), CRP and eGFR of healthy controls, kidney disease controls (KD), quiescent Systemic Lupus Erythematosus (SLE) and active SLE. GlycA is elevated in active SLE (*n* = 105), compared to healthy controls (*n* = 20), KD controls (*n* = 21), and quiescent SLE (*n* = 39) patients. CRP level is lower in quiescent SLE than in active SLE and KD controls but does not differ between active SLE and KD controls. Estimated glomerular filtration rate (eGFR) is lower in KD controls than in patients with quiescent or active SLE, thereby preventing decreased renal function from being the sole cause of the elevation of GlycA in active SLE. Significance of a Mann-Whiney-Wilcoxon test comparing the groups are indicated (* *p* < 0.05, ** *p* < 0.01, *** *p* < 0.001).

**Figure 2 jcm-09-00970-f002:**
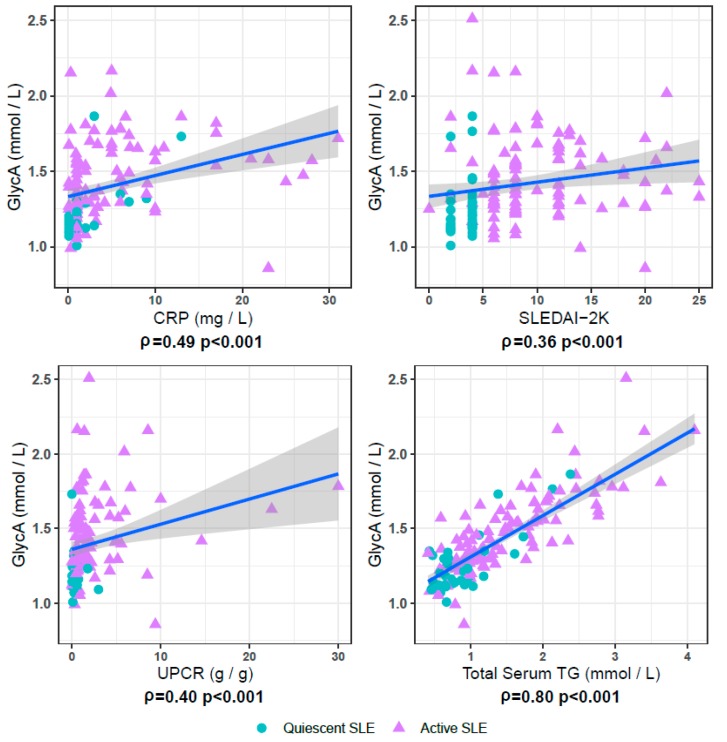
GlycA correlates with measures for disease activity in SLE patient samples. In the 144 examined SLE patient samples, comprising samples from clinically quiescent patients (blue circles, *n* = 39) and from patients with active disease (purple triangles, *n* = 105), GlycA significantly correlates with C-reactive protein (CRP), SLE disease activity index 2K (SLEDAI-2K), and urinary protein/creatinine ratio (UPCR). Also depicted is the strong correlation between serum triglyceride (TG) and GlycA levels.

**Figure 3 jcm-09-00970-f003:**
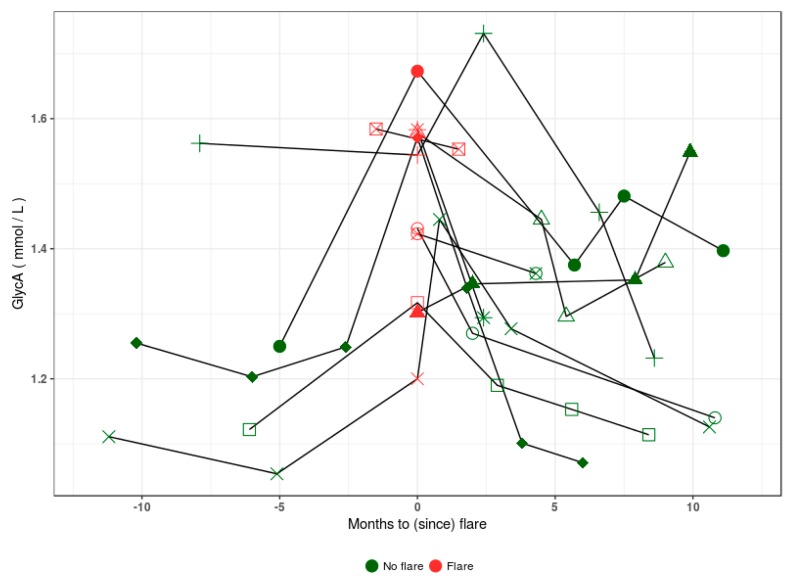
GlycA levels in longitudinal samples of 11 patients with lupus nephritis who were sampled longitudinally and at the time of a flare. In samples taken up to a year prior and after the flare event, patients show increased GlycA levels at or directly following flare presentation (red). Post flare, most patients show a return to baseline GlycA levels.

**Figure 4 jcm-09-00970-f004:**
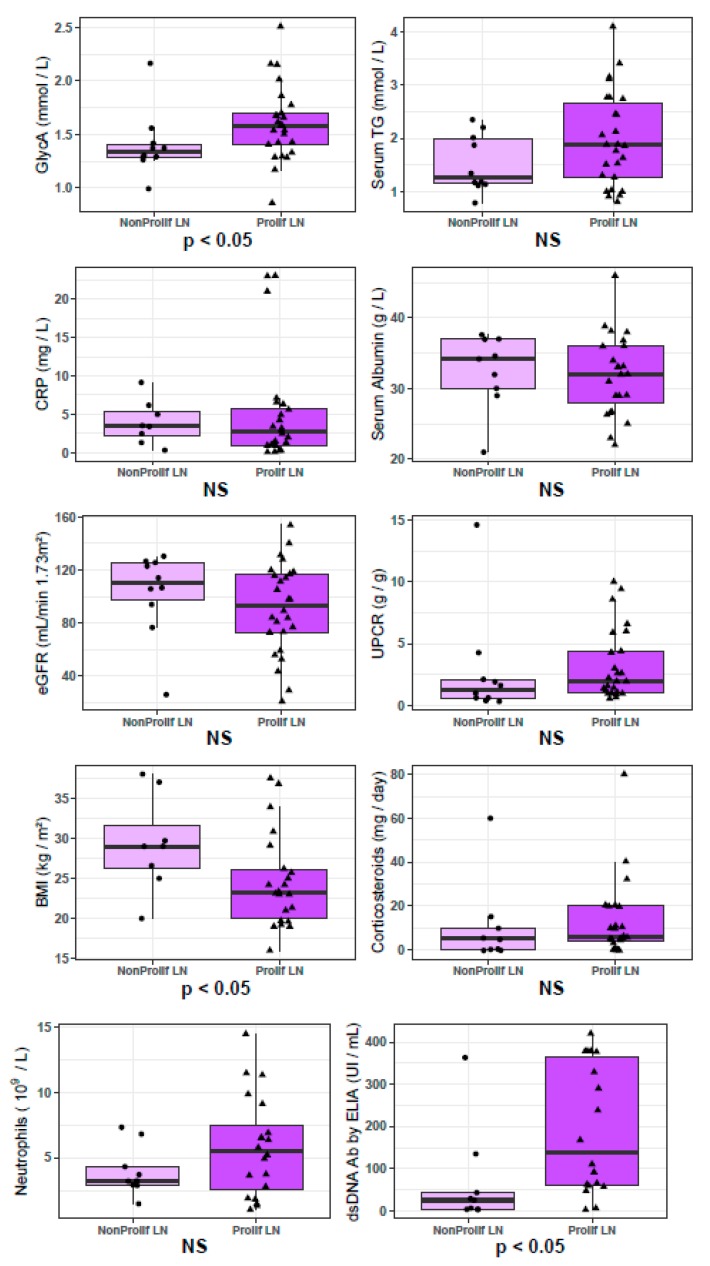
Comparison of samples taken at the time of renal biopsy, between patients with non-proliferative versus proliferative lupus nephritis. Significance of Mann-Whitney-Wilcoxon *U* tests comparing the groups are indicated below each panel. Patients with proliferative lupus nephritis (ProlifLN, *n* = 26) display significantly higher Glycoprotein Acetylation (GlycA) levels than patients with non-proliferative lupus nephritis (NonProlifLN, *n* = 26), although they have significantly lower BMI. Patients with proliferative LN also display higher levels of anti-dsDNA antibodies. Complete results are listed in Table 3.

**Figure 5 jcm-09-00970-f005:**
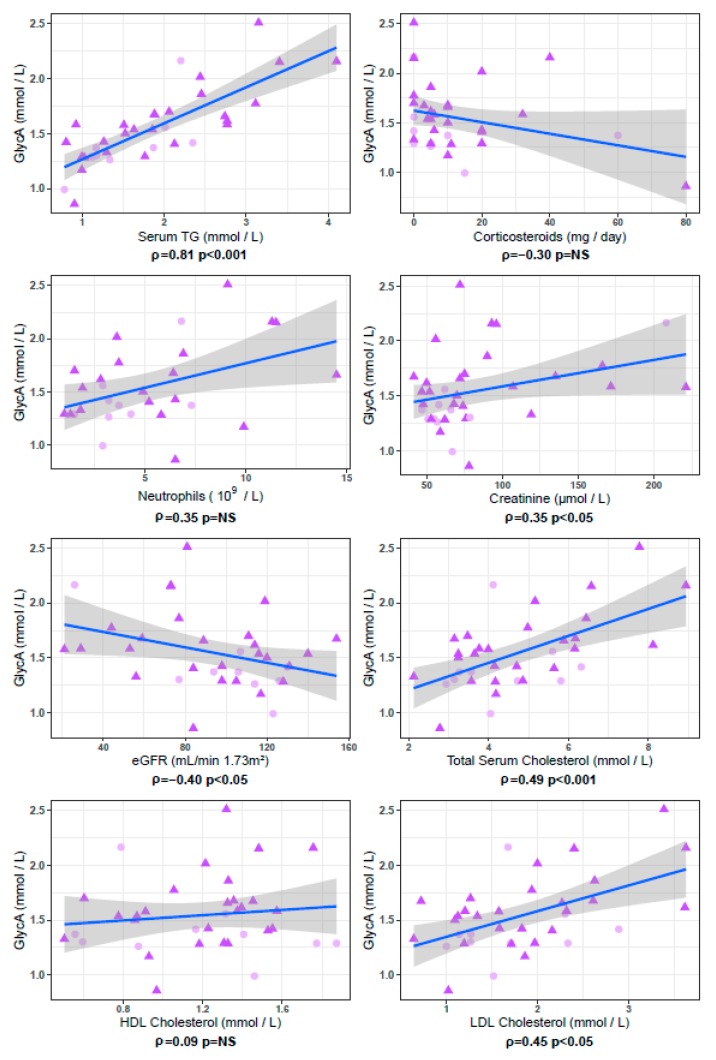
GlycA correlations in patients with lupus nephritis sampled at the time of renal biopsy. Below each panel, Spearman correlation coefficient (ρ) and *p* value are indicated. In the 36 samples collected at time of renal biopsy, originating from patients with non-proliferative (NonProlif LN, pink circles, *n* = 10) or proliferative (ProlifLN, purple triangles, *n* = 26) lupus nephritis, GlycA strongly correlates with serum TG, and correlates with total cholesterol and LDL cholesterol, but not with HDL cholesterol. GlycA shows significant association to serum creatinine levels and to estimated glomerular filtration rate (eGFR). Also depicted are two associations which do not reach statistical significance at a 0.05 threshold: a positive correlation of GlycA levels with neutrophil count (*p* = 0.06) and an inverse correlation between GlycA and daily corticosteroid dosage (*p* = 0.09). Full results are listed in Table 3.

**Figure 6 jcm-09-00970-f006:**
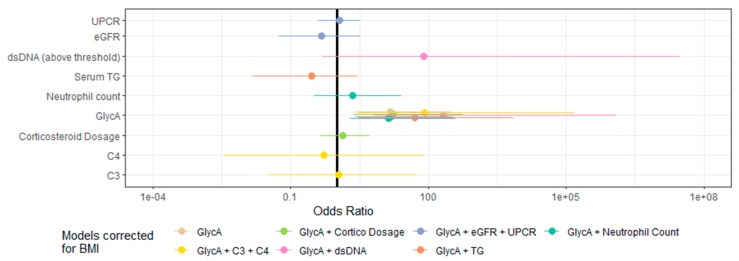
Multivariate logistic regression models for proliferative status in patients with biopsy-proven lupus nephritis. Logistic regression on samples from patients with non-proliferative (*n* = 10) and proliferative (*n* = 26) Lupus Nephritis, taken at time of renal biopsy, illustrates GlycA’s independence from other biological and immunological parameters. All multivariate models were corrected for BMI, considering its known association to GlycA and the significant difference observed between patients with proliferative and non-proliferative LN in our cohort (Table 2). Continuous variables were standardized prior to model construction.

**Table 1 jcm-09-00970-t001:** Comparison of NMR results between controls and SLE patients.

NMR Measurement	Healthy Controls (*n* = 20)	Non-Lupus Kidney Disease (*n* =21)	Quiescent SLE (*n* = 39)	Active SLE (*n* = 105)	All SLE (*n* = 144)	All SLE vs. HC	Quiescent SLE vs. HC	Active SLE vs. HC	Active SLE vs. Quiescent SLE	Active SLE vs. Non-Lupus KD
**Serum albumin** (NMR signal area)	0.089, 0.085–0.092	0.079, 0.074–0.082	0.086, 0.083–0.088	0.078, 0.070–0.081	0.079, 0.074–0.085	***	NS	***	***	NS
**Serum creatinin** (µmol/L)	51, 38–63	126, 74–218	57, 48–68	57, 47–74	57, 47–71	NS	NS	NS	NS	***
**Total cholesterol** (mmol/L)	2.48, 2.21–2.89	3.65, 3.03–5.06	4.04, 2.91–4.49	4.12, 3.53–4.92	4.08, 3.34–4.77	***	***	***	*	NS
**HDL cholesterol** (mmol/L)	0.76, 0.64–0.92	1.14, 0.99–1.26	1.55, 1.31–1.86	1.39, 1.05–1.70	1.43, 1.16–1.76	***	***	***	*	*
**LDL cholesterol** (mmol/L)	0.81, 0.68–1.11	1.41, 1.04–2.01	1.18, 0.82–1.47	1.50, 1.18–1.85	1.43, 1.11–1.71	***	*	***	***	NS
**Total triglycerides** (mmol/L)	0.89, 0.72–0.97	1.11, 0.87–1.91	0.75, 0.65–0.97	1.28, 0.95–1.91	1.10, 0.79–1.81	**	NS	***	***	NS
**GlycA** (mmol/L)	1.32, 1.14–1.45	1.29, 1.15–1.51	1.18, 1.13–1.31	1.42, 1.29–1.62	1.349, 1.232–1.564	NS	NS	**	***	*

Seven biomarkers measured by NMR were compared in a total of *n* = 194 samples. In total, five group comparisons are made: (1) Healthy Controls (*n* = 20) compared to all SLE patients (*n* = 144, comprising *n* = 39 quiescent and *n* = 105 active SLE patients), (2) HC vs. Quiescent SLE (*n* = 39), (3) HC vs. Active SLE (*n* = 105), (4) Quiescent SLE (*n* = 39) vs. active SLE (*n* = 105), and (5) Non-Lupus Kidney Disease (KD, *n* = 21) vs. active SLE (*n* = 105). Group values are listed as median and interquartile range. Significant differences between two groups were tested using the Mann-Whitney-Wilcoxon’s *U* test (* *p* < 0.05, ** *p* < 0.01, *** *p* < 0.001, red if the first group has higher values, blue if the first group has lower values). HC: healthy controls; QSLE: quiescent SLE; ASLE: active SLE; KD: non-lupus kidney disease.

**Table 2 jcm-09-00970-t002:** GlycA associations in SLE patient samples.

Characteristics	Quiescent SLE *n* = 39	Active SLE *n* = 105	All SLE *n* = 144
Age	−0.06	−0.17	−0.2
Gender female (%)	NS	NS	NS
Ethnicity caucasian (%)	NS	NS	NS
BMI (kg/m^2^)	0.07	−0.12	0.07
Current smoking (%)	NS	NS	NS
**SLEDAI-2K**	0.15	0.01	**0.36 *****
Corticosteroid daily dose (mg)	0.12	0.02	0.07
Hydroxychloroquine (%)	NS	NS	NS
**CRP (mg/L)**	**0.57 *****	**0.36 *****	**0.49 *****
**Serum albumin (g/L)**	−0.22	**−0.26 ***	**−0.41 *****
**Serum creatinin (µmol/L)**	**0.41 ***	0.25	0.16
**eGFR (mL/min 1.73m^2^)**	**−0.37 ***	−0.14	−0.13
**C3 (g/L)**	−0.06	−0.13	**−0.18 ***
C4 (g/L)	−0.08	0.04	−0.1
dsDNA antibody presence (%)	NS	NS	NS
**UPCR (g/g)**	−0.12	**0.23 ***	0.4
**Serum albumin (NMR signal area)**	0.06	−0.09	**−0.27 ****
**Serum creatinin (mmol/L)**	**0.45 ****	**0.25 ***	**0.26 ****
**Total cholesterol (mmol/L)**	0.3	**0.32 *****	**0.36 *****
**HDL cholesterol (mmol/L)**	0.05	**−0.28 ****	**−0.25 ****
**LDL cholesterol (mmol/L)**	0.24	**0.39 *****	**0.41 *****
**Total serum triglycerides (mmol/L)**	**0.43 ****	**0.81 *****	**0.80 *****

Associations of GlycA levels with patient demographic and clinical characteristics, as well as patient laboratory and NMR measurements were tested in quiescent SLE patient samples (*n* = 39), active SLE patient samples (*n* = 105) and the full combination of both groups (*n* = 144). For continuous variables, GlycA association is expressed as spearman’s correlation coefficient (ρ) and the associated *p* value is represented (* *p* < 0.05, ** *p* < 0.01, *** *p* < 0.001). GlycA levels’ association to categorical variables was tested using Mann-Whitney-Wilcoxon’s *U* test, with *p* values similarly represented.

**Table 3 jcm-09-00970-t003:** Summary statistics and GlycA association tests in patients with biopsy-proven lupus nephritis at the time of sampling.

Characteristics	Biopsy Proven Flaring Non-Proliferative LN (*n* = 10)	Biopsy Proven Flaring Proliferative LN (*n* = 26)	Available Observations	Proliferative Status Wilcoxon/Chi^2^ *p*-Value	GlycA Association
Age	35.5, 28–38.5	32, 25–39	36	0.46	−0.10
Gender Female (%)	90	88	36	1.00	NS
Ethnicity caucasian (%)	80	85	36	1.00	NS
**BMI (kg/m^2^)**	29, 26.2–31.5	23.24, 20.03–26.10	30	**0.04**	−0.002
Currently smoking (%)	40	28	35	0.77	NS
SLEDAI-2K	8, 4.5–13.5	12, 8–18	35	0.29	−0.21
Corticosteroid daily dose (mg)	5, 0–10	6, 4–20	34	0.28	−0.30
Receiving Hydroxychloroquine (%)	60	77	36	0.55	NS
CRP (mg/L)	3.5, 2.2–5.3	2.7, 0.98–5.78	32	0.60	0.01
Serum albumin (g/L)	34.2, 30–37	32, 27.9–36	32	0.60	0.07
**Serum creatinin (µmol/L)**	60, 55.5–66.75	73, 56.75–95.25	36	0.22	**0.35 ***
**eGFR (mL/min/1.73m^2^)**	110.5, 97–125.25	93.5, 73–117	36	0.28	**−0.40 ***
Neutrophil count (10^9^/L)	3.2, 2.9–4.3	5.5, 2.58–7.45	29	0.38	0.35
C3 (g/L)	1.04, 0.54–1.26	0.63, 0.45–0.88	29	0.19	–0.10
C4 (g/L)	0.22, 0.08–0.25	0.10, 0.04–0.20	29	0.57	0.13
dsDNA antibody presence (yes/no)	60	91	33	0.10	NS
**dsDNA antibody ELIA (UI/mL)**	25, 3–43	139, 60–365	27	**0.01**	−0.01
UPCR (g/g)	1.29, 0.60–2.05	1.94, 1.05–4.39	35	0.15	0.23
Hematuria (%)	67	70	29	1.00	NS
Leukocyturia (%)	56	60	29	1.00	NS
**NMR serum albumin (NMR signal area)**	0.070, 0.061–0.077	0.070, 0.066–0.079	36	0.52	**0.40 ***
**NMR serum creatinin (µmol/L)**	52, 44–55	62, 52–79	36	0.08	**0.51 ****
**Total cholesterol (mmol/L)**	4.08, 3.35–5.38	4.44, 3.59–6.09	36	0.39	**0.49 ****
HDL cholesterol (mmol/L)	1.24, 0.81–1.45	1.31, 0.94–1.39	36	0.88	0.09
**LDL cholesterol (mmol/L)**	1.598, 1.268–2.154	1.845, 1.221–2.306	36	0.64	**0.45 ****
**Serum triglycerides (mmol/L)**	1.258, 1.144–1.977	1.869, 1.273–2.669	36	0.21	**0.81 *****
**GlycA (mmol/L)**	1.336, 1.289–1.406	1.580, 1.410–1.694	36	**0.04**	**1 *****

For non-proliferative (*n* = 10) and proliferative (*n* = 26) LN samples taken at the time of renal biopsy, the association to proliferative status of all demographic and clinical characteristics, as well as laboratory and NMR measurements was tested. These characteristics were then also tested for association with serum GlycA levels. For each examined characteristic, summary statistics are expressed as median, interquartile range or percentage (for continuous variables and categorical variables, respectively) and the number of available observations is listed. Association is tested using Mann-Whitney-Wilcoxon’s *U* test or Chi^2^ test for continuous and categorical variables, as appropriate. (* *p* < 0.05, ** *p* < 0.01, *** *p* < 0.001).

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
