# Peer review of "Serum GlycA Level Is Elevated in Active Systemic Lupus Erythematosus and Correlates to Disease Activity and Lupus Nephritis Severity"

_jcm, 2020, doi:10.3390/jcm9040970_

Round 1

Reviewer 1 Report

All my concerns have been addressed adequately. I have no further comments.

Author Response

We thank again the reviewer for guiding and accompanying us in the improvement of this manuscript.

Reviewer 2 Report

The authors have answered the questions raised satisfactorily and have modified the manuscript accordingly.

I'm still concerned by the way data is presented in tables and figures:

  • Given that the statistics in this study are mostly descriptive and not adjusted for multiple comparisons, p-values should be given as 'NS' if > 0.05 and <  0.01 or <0.001 if lower than this. Notation of p-values should be held in the same manner throughout the manuscript.
  • Tables need to be simplified to show only relevant information.

E.g for table 3: no need for column with headers. No need to have 4 columns for p-values, one would suffice and statistical tests used may be noted in the figure legend. This table may be presented with 5 columns at most.

Table 2: Should be reduced to 4 columns, by indicating significant spearman’s rho values with * for <0.01 and ** for < 0.001.

Table 1. No need to give measurements with 3 decimal digits. Two decimal digit would largely suffice.

Figure 5. P = 0 does not exist. Please state p < 0.001

  • Also avoid duplicating information within the table/figure in the title/legend. This is mostly done in the manuscript already.

These additional efforts could improve readability of tables and figures. 

Author Response

We thank the reviewer for his/her thorough analysis of our work, and for guiding and accompanying us in the improvement of this manuscript. We modified the presentation of the tables and figures as suggested by the reviewer:

Q1. Given that the statistics in this study are mostly descriptive and not adjusted for multiple comparisons, p-values should be given as 'NS' if > 0.05 and <  0.01 or <0.001 if lower than this. Notation of p-values should be held in the same manner throughout the manuscript.

R1. The p-values were given as NS, <0.05, <0.01 or <0.001 throughout the manuscript.

Q2. Tables need to be simplified to show only relevant information. E.g for table 3: no need for column with headers. No need to have 4 columns for p-values, one would suffice and statistical tests used may be noted in the figure legend. This table may be presented with 5 columns at most.

R2. Table 3 was simplified as suggested.

Q3. Table 2: Should be reduced to 4 columns, by indicating significant spearman’s rho values with * for <0.01 and ** for < 0.001.

R3. Table 2 was simplified as suggested.

Q4. Table 1. No need to give measurements with 3 decimal digits. Two decimal digit would largely suffice.

R4. If the reviewer agrees, we’d rather keep Table 1 as it is, because we think reducing the decimal digits to 2 result in the following paradox: to give the same median, IQR value of 0.08, 0.07-0.08 for two groups, and then claim a significant difference between them.

Q5. Figure 5. P = 0 does not exist. Please state p < 0.001

R5. Figure 5 was modified as requested.

Q6. Also avoid duplicating information within the table/figure in the title/legend. This is mostly done in the manuscript already.

These additional efforts could improve readability of tables and figures. 

R6. We’ve removed duplicated information in the text.

This manuscript is a resubmission of an earlier submission. The following is a list of the peer review reports and author responses from that submission.

Round 1

Reviewer 1 Report

This is a well written and interesting manuscript addressing the role of GlycA as a novel biomarker for disease activity/severity in SLE and in particular LN.

The shown data are convincing and support the authors´ conclusions. The number of examined patients and controls also appears to be sufficient to answer the questions of the study. The discussion is also well balanced.

The authors might consider adding the following data in order to further enhance the validity of their observations:

  1. The authors only provide data on whether patients are positive or negative for dsDNA-Abs. It would be more informative to show a correlation between GlycA and serum levels of dsDNA-Abs determined in a quantitative way by ELISA, if available.
  2. In addition to tables it would be helpful to show also graphs for the significant correlations, in particular the correlation with SLEDAI or with other parameters referring to the main messages of the study.
  3. To complete the renal data correlation or association of GlycA with hematuria could be analysed.
  4. As biopsies from renal patients are available, the authors could analyse whether there is a correlation of GlycA levels with a quantitative histomorphological score, such as the renal activity index by Austin (Austin HA et al. Diffuse proliferative lupus nephritis: identification of specific pathologic features affecting renal outcome. Kidney Int. 1984;25(4):689-95).

Reviewer 2 Report

There is need for better biomarkers in SLE and in particular lupus nephritis.

The authors report results form a cross-sectional analysis of serum samples of patients with systemic lupus erythematosus (SLE), patients with other kidney diseases and healthy controls. They show that serum glycoprotein acetylation (GlycA) assessed by nuclear magnetic resonance spectroscopy is increased in active SLE and in particular in patients with active proliferative lupus nephritis. They clearly show that GlycA levels are increased in active proliferative lupus nephritis (LN). However, in the few patients with LN that had longitudinal GlycA assessement, levels do not appear to rise in the months prior to overt LN. Also in contrast to previous studies, GlycA levels are not increased in the SLE group as a whole compared to controls.

GlcA appears of interest as marker of active proliferative lupus nephritis, but deserves further studies. Given it's strong correlation with triglycerides shown in this study it may also be of interest as a marker for accelerated atherosclerosis, which is a major issue in SLE patient care.

The manuscript is well written.

Major comments: 

1. The 21 non-lupus-related kidney disease where neither matched for age nor sex to SLE patients, in addition to being more impaired in renal function, so this may limit comparison of GlycA levels between these groups. Also I was unable to find information on age and sex distribution of the 29 healthy controls. This information should be given.

2. I'm not confident with tables 1 + 2 showing only p-values. The interesting stuff is in the supplementary tables. I suggest to combine the content of the supplementary tables with the statistics as main tables and highlight only the differences that are significant.

3. As the authors point out CRP is not a reliable activity marker in SLE. I miss the correlation of GlycA with erythrocyte sedimentation rate (ESR), which is a very inexpensive and quite reliable marker of disease activity in SLE, although of poor specificity. Are ESR mesurements available from the patients included in this study ? 

4. The positive correlation of serum GlycA levels and triglycerides is interesting. Did the authors look whether there was also an association between triglycerides and BMI observed in patients with proliferative lupus nephritis ?

Minor comments:

Table/Figure titles may be improved by indicating the number of patients featured. Some typos (e.g. in table 2).